# Secret Recipe Revealed: Chemical Evaluation of Raw Colouring Mixtures from Early 19th Century Moravia

**DOI:** 10.3390/molecules27165205

**Published:** 2022-08-15

**Authors:** Klára Jagošová, Martin Moník, Jaroslav Kapusta, Radka Pechancová, Jana Nádvorníková, Pavel Fojtík, Ondřej Kurka, Tereza Závodná, Petr Bednář, Lukáš Richtera, Lukáš Kučera

**Affiliations:** 1Department of Analytical Chemistry, Faculty of Science, Palacký University, 17. listopadu 12, 77900 Olomouc, Czech Republic; 2Department of Geology, Faculty of Science, Palacký University, 17. listopadu 12, 77900 Olomouc, Czech Republic; 3Institute of Archaeological Heritage Brno, v.v.i.-workplace Prostějov, Tetín 8, 79601 Prostějov, Czech Republic; 4Department of Chemistry and Biochemistry, Mendel University in Brno, Zemědělská 1, 61300 Brno, Czech Republic

**Keywords:** Faience, direct mass spectrometry, elemental analysis, inorganic pigments, mass spectrometry, organic additives

## Abstract

An archaeological excavation in Prostějov (Czech Republic) revealed a workshop of a local potter with colourless, pink, and blue powders presumably used to produce faience/surface decoration. A comprehensive analytical study, which combined elemental and molecular analysis techniques, was performed to shed light on the chemical composition of these unique findings. Scanning electron microscopy with energy dispersive X-ray spectroscopy (SEM EDX), inductively coupled-plasma mass spectrometry (ICP MS), flow injection analysis (FIA) with electrospray ionisation mass spectrometry (ESI MS), laser desorption ionisation mass spectrometry (LDI MS), and Raman spectroscopy were applied to reveal the elemental composition of the powders and identify the colouring agents in the pink and blue powders. The colouring agents in the pink powder were probably iron and the agent in the blue powder is Prussian blue. On top of that, it was also possible to determine the organic additives in these powders through pyrolysis gas chromatography with mass spectrometric detection (Py GC/MS), atmospheric solids analysis probe ion mobility mass spectrometry (ASAP IM MS), and LDI MS. The organic constituents were identified as plant resin, beeswax, and fats. These results point to the preparation of faience/pigment mixtures as oil paint.

## 1. Introduction

The chemical analysis of glazed pottery findings remains a big challenge because of usually minute amounts of unique samples available for analysis, the complexity of the matrix, and the transformation of the individual components during manufacturing processes. Many comprehensive studies on archaeological glazed pottery have been carried out combining microinvasive and non-invasive analytical techniques [1,2,3,4,5,6,7,8,9,10]. The main goal of these studies was to reveal the composition of the ceramic matrix, determine the colouring agents and additives in the glazed layers (faience), reconstruct the manufacturing technology, and estimate the provenance of the pottery. Electron microscopy combined with energy dispersive X-ray spectroscopy (SEM-EDX) and X-ray fluorescence spectrometry are often the methods of choice for the determination of elemental composition. The disadvantage of SEM-EDX and XRF is their lower sensitivity. Due to this limitation, ICP-MS is preferred when the pattern of trace elements needs to be determined. For the analysis of cross-sections, it is advantageous to use laser ablation coupled with ICP-MS since it has a suitable resolution (i.e., tens of micrometres) [3,6,11,12,13]. Among other techniques of elemental analysis, thermal ionisation mass spectrometry [7,8], secondary ion mass spectrometry [14], particle induced X-ray or gamma ray spectroscopy [4,5], X-ray diffraction [1,9], laser induced breakdown spectrometry [15], neutron activation analysis [16] or X-ray photoelectron spectrometry [17] have been applied for glaze/faience analyses. Methods of molecular spectroscopy, such as Raman spectroscopy or infrared spectroscopy, are often utilised for the determination of the mineralogical structure of studied materials [18,19]. Until the present day, no study revealed the complete manufacturing technologies of faience production since only final products or fragments are available for physicochemical analysis using modern analytical techniques. The reason is the thermal degradation of organic additives during the firing of faience on ceramics. It should be noted that Zaremba et al. [20] observed the emission of CO_2_ and ammonia during the thermal analysis of the glazed and underglazed layer of the faience Ptolemaic bowl. They assumed that the evolved gases were decomposition products of organic matter trapped in the pores of the object. However, even in this case, it was not possible to determine the original organic additives.

The technique of faience was applied to Moravian pottery (Czech Republic) from the 16th century onwards [21,22]. It consists of applying a glassy layer on fired sherd to make it harder, impenetrable, and easier to paint on. It was brought to the Czech lands by Anabaptists in the 16th century, probably from southern Germany, and remained in use after their expulsion from Bohemia and Moravia (1622 AD) after the Battle of White Mountain 1620 AD [21]. In 2017, a rescue archaeological excavation in Vodní ulice Street (*Wassergasse*) in Prostějov (Moravia, Czech Republic) discovered the house of pottery master Jan Skřivánek, born in 1773 and active in Prostějov after 1802 [23]. Three powder samples were found inside preparative ceramic vessels, i.e., colourless/white (sample 1), pink (sample 2), and blue (sample 3) powders (Figure 1).

The aim of the presented study is to identify the inorganic and organic components of the powders since their recipes were kept secret and formed a part of the “arcane” knowledge of folk alchemists [21]. Although a number of recipes for such powders are known to historians, ibid., these were also often written in “secret” language so that no other pottery master could make use of them. In this way, this study helps to decode this language and understand the beginning part of the manufacturing process of faience. Untargeted analyses of powders were performed by ASAP-MS with ion mobility (IM), FIA/ESI-MS, Py-GC/MS, LDI-MS, and Raman spectroscopy. For elemental composition, SEM-EDX and semiquantitative ICP-MS analyses were performed. To the best of our knowledge, this is the first-time the analysis of raw faience materials has been conducted by modern instrumental analytical techniques.

## 2. Results

The morphological characterisation of inorganic grains of particular minerals was performed using SEM/EDX (Table 1). Based on the morphology of fragments observed with SEM and their chemical composition, the colourless powder is formed by crushed glass, one of the components of a so-called *frit* (Figure 2a) [24]. In practice, it is later mixed with water and colourants and applied to the pottery surface before the second firing. Some fragments are enriched in Pb (EDX analyses 1, 2, and 4 in Table 1 and Figure 2a,b), ranging from 6 wt% to 18 wt% of PbO while others (analysis 3) are almost pure silica. Sulphur is present in small quantities (<0.3 wt% of SO_3_) in all analysed samples, probably linked with the Pb mineral (galena). A variability in chemical composition within single pieces of crushed glass/frit was observed (Figure 2b), indicating the mixing of several types of glass. Notably, a high amount of lead in various forms was confirmed by LDI-MS as well. The signals of clusters with different elemental compositions dominate the LDI-MS spectra in both ionisation modes (see Appendix A).

The pink-coloured powder turned out to be very fine-grained (Figure 2c) so only “bulk” SEM-EDX analysis was performed. Apart from the predominant SiO_2_ and Al_2_O_3_ oxides, we encountered CaO, PbO, and K_2_O in the mixture; (hydr)oxides of P_2_O_5_, FeO, TiO_2_, MgO, and Na_2_O, together with SO_3_, were present in amounts < 1 wt%. The blue powder comprises abundant baryte crystals (EDX analyses), a common additive in faience manufacture which served as flux, increased the brilliancy of the glaze, and reduced the number of bubbles during melting [24,25]. The remaining matter also contains SiO_2_, Al_2_O_3_, PbO, and FeO, and in lesser or trace amounts CaO, P_2_O_5_, MgO, SrO, or Na_2_O as well. With regard to the manufacturing technology, the addition of Pb both in the crushed glass and in the two coloured powders is obvious. The slightly elevated values of K_2_O and Na_2_O indicate that we have mixtures for lead glazes with low amounts of alkalis [3]. Though elevated, the amounts of Pb (wt% from SEM-EDS) in the powders from Prostějov are still much lower than in Modern Age glazes published by Tite et al. or Gregerová et al. [3,22]; therefore, it can be assumed that another Pb-rich mixture, not found during excavation, was added. Lead was probably available in the form of litharge (PbO), either a secondary mineral from outcrops of galena ores, or a by-product from smelting (lead-) silver ores [26]. Similar to baryte, litharge served as a flux in glazes, increasing their brilliance, smoothness, density, and resistance to cracking [3,14]. Its disadvantage was the release of lead in meals, resulting in higher Pb-intake by the early Modern Age population, with potentially poisonous effects [27,28]. Most of the analysed mixtures, especially the pink powder and crushed glass, contain small amounts of P_2_O_5_, CaO, and MgO, but their amounts are lower than in the Anabaptist pottery from Strachotín (Moravia) where the use of bone ash from calcined bones or antlers was conjectured [22]. As it stands, the mixtures from Prostějov would probably result in translucent rather than opacified glazes. Only the joint presence of TiO_2_, SO_3_, and FeO in the pink powder may potentially indicate the use of titanium oxide opacifiers, manufactured by soaking ilmenite (FeTiSiO_3_) in green vitriol, i.e., ferrous sulphate (FeSO_4_).

In addition to SEM-EDX, ICP-MS analyses were performed to obtain an exact concentration of elements in the studied powders and reveal the colour origin (Table 2). One of the most common red pigments used in glazes/faience was hematite α-Fe_2_O_3_ (or maghemite γ-Fe_2_O_3_) which causes red to brown shades depending on the conditions during firing. The red hue was achieved under oxidising conditions at a temperature higher than 700 °C [29]. Red glazes were also achieved by the use of other minerals, for example, cuprite Cu_2_O, crocoite (PbCrO_4_), or red lead (Pb_3_O_4_) [29,30,31]. Based on the results of the ICP-MS analysis, though, we hypothesise that the colouring agent in the pink powder was Fe found in concentrations of 0.50 mg·g^−1^, probably in the form of hematite. Copper, e.g., in the form of cuprite or copper nanoparticles, was also used as a red colourant in historical glassmaking [32,33], but the amount of Cu used in such cases was about 100x greater than in the red powder from Prostějov (0.26 mg·g^−1^, Table 2). 

Py-GC/MS analysis revealed the presence of organic additives in the pink faience powder, i.e., diterpene resin acids of the abietane and pimarane types (Figure 3a). Abietic acid, dehydroabietic acid, pimaric acid, and levopimaric acid are significant components of resins produced by conifers (gymnosperms), e.g., pine (*Pinus*), larch (*Larix*), spruce (*Pices*), and fir (*Abies*). [34,35]. ASAP-IM-MS did not allow the separation of individual resin acid isomers but confirmed the presence of the resin as a sum of all isomers. A compound with m/z 301.2145 (C_20_H_29_O_2_^+^) corresponds to the structure of protonated dehydroabietic acid with the deviation from theoretical mass (dtm) of −2.3 mDa. The fragmentation spectra of dehydroabietic acid in the pink powder and in the standard are presented in Figure 3b,c, respectively. In addition to the signals belonging to the resin, the signals at m/z 463.4858, 491.5186, and 519.5491 correspond to the ions of molecular formula C_32_H_63_O_1_^+^, C_34_H_67_O_1_^+^, C_36_H_71_O_1_^+^ with dtm −2.1, −0.6 and −1.4 mDa, respectively (for fragmentation spectra see Appendix A). Those signals were assigned as thermal/ion fragments of beeswax components based on a comparison with the mass spectrum of a standard of beeswax. As further evidence of beeswax presence, we also identified molecule C_42_H_83_O_2_^+^ (619.6393 Da) with dtm −0.8 mDa as an ester of palmitic acid, which is a typical component of beeswax [36] (for fragmentation spectra see Appendix A). In addition, diglycerols also were found with m/z 607.5640 (C_39_H_75_O_4_^+^; dtm −2.5 mDa; distearoyl glycerol, SS), 579.5327 (C_37_H_71_O_4_^+^; dtm −2.5 mDa; palmitoyl-stearoyl glycerol; PS), and 551.4991 (C_35_H_67_O_4_^+^; dmt −4.8 dipalmitoyl glycerol, PP). (for fragmentation spectra see Appendix A). Those compounds are present in high concentrations in fatty materials (originating from both plants and animals) [36]. The pink powder was probably prepared in a similar way as pigments for traditional oil paintings. According to Slánský, the preparation of traditional oil paints includes pigment(s) and oil as main components with the frequent presence of resin, wax, or balms [37].

The blue colour in glaze making is normally achieved by the addition of Cr, Co, or Cu [11,26,38], but none are present in the powder from Prostějov–Vodní ulice Street. According to the ICP-MS results, the bulk of the blue powder contained an increased amount of iron (approximately 4 mg·g^−1^) which could be the source of the blue colour as hexacyanoferrate pigment Fe_4_[Fe(CN)_6_]_3_, commonly known as Prussian blue. The presence of PB was consequently identified and proved by the means of FIA/ESI-MS and Raman spectroscopy. The FIA/ESI-MS spectrum of PB standard contained the most intense signals of [Fe(CN)_3_]^−^ at m/z 133.9443 Da (Figure 4a). In the spectrum of the blue powder, this signal was found as well (m/z 133.9434), with dtm −0.8 mDa (Figure 4b). These results were consequently supported by Raman spectra of the reference material and sample (see Figure 4c). Four significant signals were found in Raman spectra, i.e., 2154, 2090, 527, and 275 cm^−1^. The vibration at 2154 cm^−1^ corresponds to ν(C≡N) stretching vibration and [Fe(II), Fe(III)] vibrational state. This peak is followed by a shoulder of a characteristic CN^−^ vibration, and signal at 2090 cm^−1^ that corresponds to the ν(C≡N) stretching vibration of the [Fe(II), Fe(III)] state. The remaining two signals at 527 and 275 cm^−1^ refer to Fe–C stretching vibrations of the lattice and Fe–CN–Fe bond deformation vibrations, respectively [39]. Note that the high concentration of sulphur, which was found in the sample of blue powder by SEM-EDX and ICP-MS (Table 1 and Table 2), points to the traditional preparation of PB, i.e., the usage of green vitriol (FeSO_4_·7 H_2_O) and dried cattle blood as the source of cyano or/and ferrocyanide groups [29]. The problem with PB, though, is that it would probably not have survived the sintering process in glaze melts. In oxidising conditions, the ferrocyanide bonds would have broken and disassembled (and likely oxidised) into Fe^3+^. We assume that in reduction conditions, PB possibly formed Fe^2+^ which can cause the aqua blue colour of the glaze. [29]. It would seem that the potter from Prostějov manufactured a mixture based on Prussian blue which he ultimately did not achieve but was left with a perfectly viable alternative. Throughout the 18th century, PB was not used for pottery glazes from other parts of the world either, as it was mostly used for paintings [26]. According to Černohorský, the blue powder could be utilised for decorative purposes after the second firing. This decorative technique was well known in the Moravia region since approximately the 1760s and was known under the German term Űberglasumalerei [21]. In the findings of Anabaptist faience from 17th century Hungary, another blue colourant based on CoO+As_2_O_3_, FeO, and NiO was used [1]. This was applied on tin-based glazes whereas Prostějov potter Jan Skřivánek either manufactured lead glazes (see Table 1) or his tin mixtures were not found during excavation.

Analysis of the organic binders in blue powder was performed by ASAP-IM-MS and LDI-MS. The content of organic additives was substantially lower in this sample compared to the sample of the pink powder. In the blue powder, we identified only the same diglycerols as in the case of the pink powder, SS, SP, and PP with dtm 1.2, 1.2, and 1.3 mDa, respectively (for fragmentation spectra see Appendix A). The LDI-MS in the negative ionisation mode detected three common fatty acids, i.e., palmitic, stearic, and oleic acid in the blue faience powder, further confirming the presence of fatty material in the faience materials (see Appendix A). We hypothesise that the blue mixture also was prepared similarly to oil paints. None of the aforementioned compounds were found in colourless powder.

## 3. Materials and Methods

### 3.1. Chemicals

Abietic acid (Lachema, Brno, Czech Republic), acetone (HPLC grade ≥ 99.8%, Fisher Scientific, Waltham, MA, USA), acetonitrile (Honeywell, Charlotte, NC, USA), dehydroabietic acid (Sigma Aldrich, St. Louis, MI, USA), hydrochloric acid (Analpure, 34–37%, Analytika, Praha, Czech Republic), hydrofluoric acid (Analpure, 48%), leucine enkephalin (HPLC ≥ 95%, Sigma Aldrich), methanol (LC-MS ≥ 99.9%, Honeywell), Mili-Q water (Merc), nitric acid (Analpure, 67–69%), N,O-Bis(trimethylsilyl)trifluoroacetamide (BSTFA) (≥99.0%, Sigma Aldrich), Prussian blue (laboratory prepared), red phosphorus (Sigma Aldrich) for the mass calibration of LDI-MS), sodium formate for the TOF calibration (prepared by mixing 100 µL 0.1 M sodium hydroxide (Fluka, Buchs, Switzerland) with 200 µL 10% formic acid (99–100%, Analar Normapur) and diluted with a mixture of acetonitrile/water (80:20, *v*/*v*), and sodium hydroxide (Penta, p.a., Brno, Czech Republic) were used.

### 3.2. Scanning Electron Microscopy–Energy Dispersive X-ray Spectrometry (SEM/EDX)

Raw faience materials were mounted on carbon foil, carbon coating (25 nm), analysed with SEM-EDX (JEOL JXA-8600), and the chemical composition of single minerals or fragments of glass were established. In other instances, and in the case of the very fine-grained pink powder, “bulk” analyses were performed by targeting a larger area to determine the composition of the whole mixture. The conditions of analysis were set as follows: accelerating voltage 15 kV, beam current 10 nA, beam diameter 1–5 µm, and counting time 60 s per spectrum. Acquired spectra were quantified by IdFix software (remX GmbH) and the following set of standards: albite (Na), diopside (Mg, Ca), microcline (Si, Al, K), apatite (P), barite (Ba, S), ilmenite (Ti, Fe), strontianite (Sr), and lead metal (Pb).

### 3.3. Inductively Coupled Plasma–Mass Spectrometry (ICP-MS)

A microwave digestion unit MLS 1200 Mega (Milestone, Italy) was employed for sample mineralisation. Prior to the mineralisation step, the powder samples were manually milled and homogenised. Subsequently, approximately 20 mg of homogenised samples were digested with a mixture of concentrated HNO_3_ (1 mL), HCl (3 mL), and HF (0.125 mL) according to a seven-step digestion program consisting of 2 min at 250 W, 2 min at 0 W, 5 min at 400 W, 2 min at 0 W, 2 min at 500 W, 2 min at 0 W, and 6 min at 600 W. After digestion and cooling, the samples were diluted with deionised water to 10 mL, transferred into polypropylene tubes and stored at 4 °C until ICP-MS analysis. Blank samples were prepared by digestion of the mixture of acids but without the presence of the sample.

All measurements were carried out using an inductively coupled plasma mass spectrometer 7700x ICP-MS (Agilent Technologies, Tokyo, Japan), fitted with an ASX-520 autosampler, a MicroMist concentric nebuliser, a cooled Scott-type double pass spray chamber, and an octopole reaction cell operating in Helium mode to overcome spectral interferences. The instrumental conditions for the semi-quantitative analysis of selected isotopes (up to 70 elements) by ICP-MS were set as follows: RF power of 1600 W, plasma gas Ar flow rate of 15.0 L·min^−1^, an auxiliary gas Ar flow rate of 0.61 L·min^−1^, nebuliser gas Ar flow rate of 0.36 L·min^−1^, collision gas He flow rate of 5 mL·min^−1^ (He mode) or 0 mL·min^−1^ (No Gas mode). All ICP-MS analyses were performed in six replicates. Obtained data were evaluated in MassHunter (Agilent Technologies, Palo Alto, CA, USA).

### 3.4. Pyrolysis–Gas Chromatography/Mass Spectrometry (Py-GC/MS)

Gas chromatographic analysis was carried out on an Agilent 8890 coupled to a mass detector Agilent 5977B (Agilent, Santa Clara, CA, USA). Pyrolysis of samples was performed with a multi-shot pyrolyzer EGA/PY-3030D (Frontier Lab, New Ulm, MN, USA) at 500 °C for 0.5 min. A capillary column Ultra Alloy UA5(MS/HT), 30 m × 0.25 mm × 0.25 μm (Frontier Laboratories Ltd., Fukushima, Japan) was operated with helium as a carrier gas at a constant flow rate of 1 mL·min^−1^. The oven program was as follows: 50 °C for 4 min followed by a 20 °C/min ramp to a final temperature of 320 °C and held for 15 min. The total run time was 32.50 min. The injection was performed in a split mode with a 60:1 split ratio. MassHunter Qualitative Analysis software (Agilent Technologies, Palo Alto, CA, USA) was used for data evaluation. A microsample of the potter’s pink powder was put into a pyrolyzer cup. Derivatisation was carried out with 2 μL of BSTFA prior to Py-GC/MS analysis.

### 3.5. Direct Mass-Spectrometric Analysis

ASAP-IM-MS and FIA/ESI-MS analyses were performed on Synapt G2-S (Waters, Milford, MA, USA) equipped with an atmospheric solid analysis probe (ASAP) and electrospray ionisation (ESI) source, a hybrid QqTOF mass analyser, and an ion mobility cell (TriWave). Exact masses of analytes were obtained by correction to the mass of leucine enkephalin (556.2771 Da) at a concentration of 2 ng·mL^−1^ in water/ACN 80/20 (*v*/*v*) with 0.1% formic acid, which was measured during each run. Data acquisition and processing were performed by MassLynx (Waters, Manchester, UK). LDI-MS analyses were done using a high-resolution tandem mass spectrometer Synapt G1 equipped with a Q-TOF analyser and vacuum MALDI ion source (Waters). Obtained spectra were treated using MassLynx 4.2. In all cases, the TOF analyser was set to resolution mode (V-mode).

#### 3.5.1. Atmospheric Solids Analysis Probe Ion Mobility Mass Spectrometry (ASAP-IM-MS)

Analyses of the three powders were performed by means of ASAP-IM-MS. This ionisation technique utilises a stream of hot nitrogen to evaporate analytes deposited on a glass capillary tube and ionises them in corona discharge [40]. Since this approach does not require extensive sample preparation, samples were prepared in a form of acetone suspension (5 mg·mL^−1^) that was introduced into an open-ended glass capillary by capillary elevation. After evaporation of the solvent, roughly 10 µg of the solid sample was subjected to analysis. The glass capillary was transferred to the mass spectrometer ion source and the analytes were evaporated by a stream of nitrogen and finally reached the temperature of 600 °C. Evaporated analytes were subsequently ionised in the plasma created by a corona discharge, and the ionised species were additionally separated according to their shape, size, and charge in the ion mobility cell. The experimental conditions were set as follows: Ionisation mode positive, analysis time 3 min, trap collision energy 4 eV, transfer collision energy 2 eV, source temperature 100 °C, sampling cone 80 V, corona current 2 µA, nitrogen flow rate 500 L·h^−1^, nitrogen temperature of 500–600 °C, IM wave velocity of 550 m·s^−1^, IM wave height of 40 V, and helium drift gas. The parameters for the fragmentation experiments were as follows: LM resolution 15, and transfer collision energy (transfer CE) values are listed in Appendix A for each compound.

#### 3.5.2. Flow Injection Analysis Electrospray Ionisation Mass Spectrometry (FIA/ESI-MS)

FIA/ESI-MS was utilised for the analysis of the decomposed blue pigment in an alkaline solution. Sample preparation and method parameters were adopted from [41]. The parameters were set as follows: Ionisation mode negative, capillary voltage 2.30 kV, sampling cone 30 V, analysis time 2 min, trap collision energy 4 V, transfer collision energy 2 V, source temperature: 120 °C, desolvation gas 250 °C at a rate of 360 L·h^−1^, nebuliser 6 bar, mass range 50–600 Da, and scan time 1 s. A mixture of methanol and water (1:1, *v*/*v*) at a flow rate of 0.075 mL·min^−1^ was used as a mobile phase, and 5 µL of the sample was injected.

The sample of standard Prussian Blue (PB, synthesised in our laboratory by mixing a solution of 1 M iron(III) chloride with a 1 M solution of a hexacyanoferrate(II) salt) was prepared at a concentration of 0.05 mg·mL^−1^ in 0.04 M solution of NaOH. The decomposition was supported by sonification in an ultrasound bath at laboratory temperature in sweep mode for 30 min. The blue powder sample was decomposed in the same way, i.e., approximately 2.5 mg were weighed and dissolved in 10 mL of 0.04 M NaOH. A blank sample was prepared from a 0.04 M solution of NaOH. All samples were diluted ten times and filtered with syringe filters before injection into the system.

#### 3.5.3. Laser Desorption-Ionisation Mass Spectrometry (LDI-MS)

Raw faience materials were fixed on a standard stainless steel MALDI plate using double-sided tape (Ulith, Prague, Czech Republic) and the MALDI plates were inserted into the MALDI chamber for analysis. Mass calibration for the LDI-MS experiments was done by the measurement of red phosphorus which was deposited as acetone suspension (10 mg·mL^−1^) on a selected position on the MALDI plate. Samples were analysed in both positive and negative ionisation modes without the application of a matrix. Parameters of LDI-MS were set as follows: sample plate 10 V, hexapole bias 10, source gas flow 0 mL·min^−1^, trap collision energy 6 eV, cooling gas flow 10 mL·min^−1^, transfer collision energy 4 eV, laser energy 500 (arb), trap gas flow 1.5 mL·min^−1^, and laser firing rate 200. Manual laser aiming was used.

### 3.6. Raman Spectroscopy

Raw faience materials were analysed using a DXR2 Raman microscope (Thermo Scientific, Waltham, MA, USA). The parameters were as follows: laser wavelength 785 nm, laser power 1 mW, aperture 50 mm slit, collect exposure time 2 s, and 16 sample exposures 16. Omnic 9 (Thermo Scientific, Waltham, MA, USA) was used for data evaluation.

## 4. Conclusions

Modern analytical techniques were used to evaluate historical raw faience materials. The colouring agent in the pink powder was probably iron in the form of Fe_2_O_3_. The colourant in blue powder is Prussian blue. To the best of our knowledge, this is the first evidence of PB in faience materials. Additionally, Py-GC/MS and ASAP-IM-MS determined constituents of plant resin, beeswax, and fats in the pink powder. The blue powder contained only fat constituents (according to ASAP-IM-MS and LDI-MS). Both coloured powders were probably prepared in a manner also used for the preparation of oil paints [37]. The colourless powder consists of various glass materials, lead as the glaze-forming ingredient, and possibly opacifiers. An obvious difference is observable between the technology used by Hungarian Anabaptists from the 17th century [1] and the potter from early 19th century Prostějov, especially regarding the glazes and blue colour, but it is possible that some parts of the glaze mixture from Prostějov are missing. Some minerals present in the potter’s powders, or crushed glass, must have been imported from other parts of Moravia, Silesia, Bohemia, or elsewhere. Baryte (BaSO_4_), the source of Ba in the blue powder, is known from several polymetallic ore deposits in the Bohemian Massif such as Horní Benešov, Zlaté Hory (then Zuckmantel) in Silesia (83 km from Prostějov [42]) or the suburbs of Jihlava (110 km) in the Bohemian-Moravian Highlands. The latter two sources comprise galena mineralisation [42,43], a possible source of Pb in the pink and blue powders and the crushed glass. As major mining activities around Jihlava occurred in the Middle Ages, the deposits from Zlaté Hory (or elsewhere) are more likely to be the sources [42] to have supplied the pottery masters in Prostějov. This seems more likely, as Zlaté Hory was a traditional source of chalcopyrite and other Cu-ores [44], potential colourants of the analysed pink powder. Apart from Zlaté Hory, a possible source of the lead (litharge) is the deposit at Příbram-Zlaté hory, a major source of Central European lead and silver in the 19th century [45], albeit somewhat distant from Prostějov (225 km). If green vitriol (FeSO_4_) was indeed used in the glaze mixtures, it could have been acquired from any pit water of the mentioned deposits (i.e., Zlaté Hory or another). To the best of our knowledge, this is the first study describing the composition of raw faience materials by modern analytical techniques and the obtained results help to understand the early part of the manufacturing process in the Central European region.

## Figures and Tables

**Figure 1 molecules-27-05205-f001:**
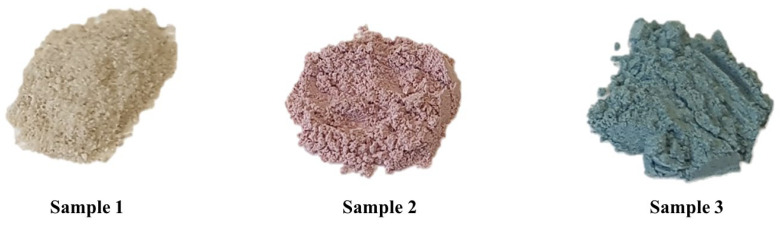
From the left: colourless, pink, and blue powders.

**Figure 2 molecules-27-05205-f002:**
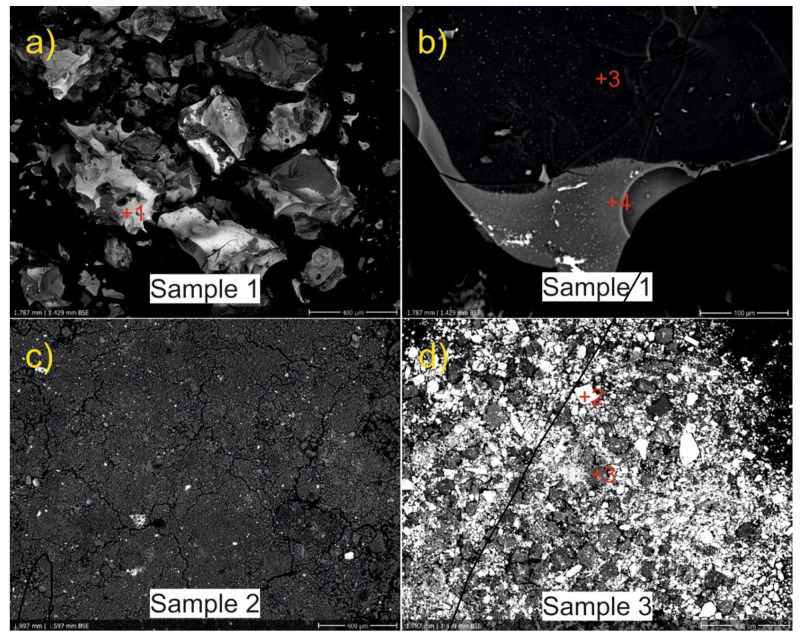
BSE images of the powders encountered in Vodní ulice Street. Crushed glass or frit (**a**,**b**) composed of glass of variable chemistry, pink powder (**c**), and blue powder (**d**). The numbers in red indicate the points of SEM-EDX analysis and correspond to the numbers of analyses in Table 1.

**Figure 3 molecules-27-05205-f003:**
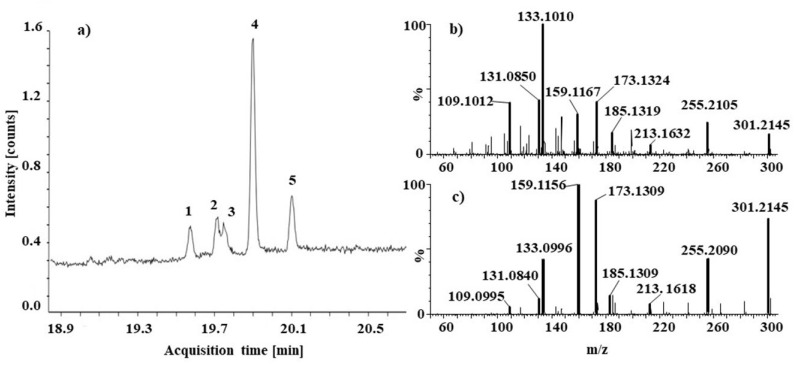
Pyrogram of the pink powder (1: pimaric acid TMS, 2: isopimaric acid TMS, 3: unknown compound, 4: dehydroabietic acid TMS, and 5: abietic acid TMS) (**a**); ASAP IM MS fragmentation spectra of compound m/z 301.2152 in pink powder (collision energy 20 V) (**b**) and in the standard of dehydroabietic acid (collision energy 20 V) (**c**).

**Figure 4 molecules-27-05205-f004:**
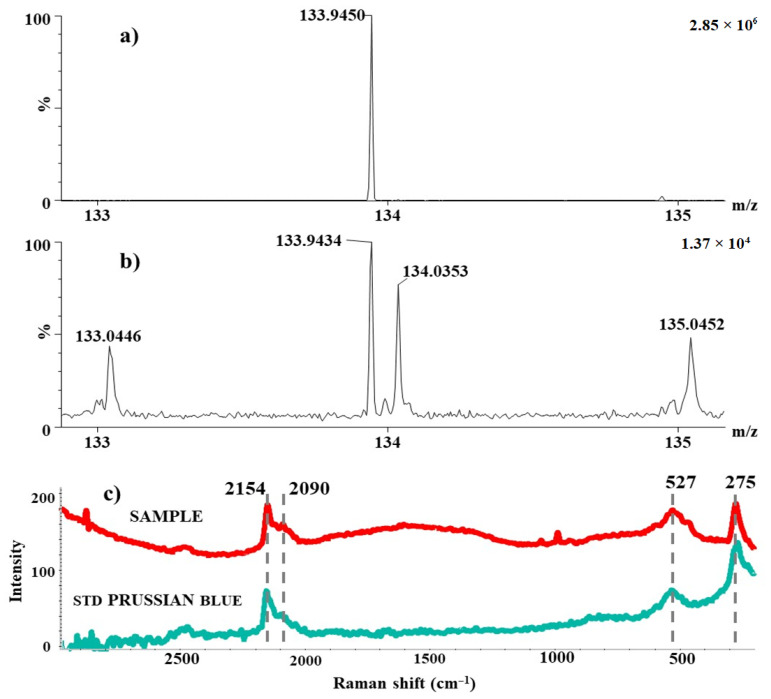
MS spectrum of the PB standard (**a**), MS spectrum of analysis of the blue powder (**b**), Raman spectrum of the blue powder, and the PB standard (**c**).

**Table 1 molecules-27-05205-t001:** EDX results of the three samples in wt%.

No. of Analysis	Sample	Na	Mg	Al	Si	P	S	K	Ca	Ti	Fe	Sr	Ba	Pb	Sum	Other Minerals
1	#1	0.98	0.17	8.52	65.21	0.05	0.22	2.42	0.22	ND	0.94	ND	0.27	11.76	90.76	quartz
2	#1	0.85	0.27	6.40	66.54	0.05	0.15	1.87	0.27	ND	1.30	ND	ND	18.13	95.82	
3	#1	0.36	0.28	0.62	101.83	0.42	0.43	0.11	0.05	ND	0.05	ND	ND	0.13	104.26	
4	#1	1.37	0.69	9.21	75.92	0.54	0.65	2.16	0.33	1.45	1.70	ND	ND	6.59	100.62	
1 (bulk)	#2	0.26	0.60	20.95	38.58	1.39	1.31	1.69	4.21	0.69	0.60	ND	ND	1.97	72.24	baryte, Na-feldspar, chlorite, and aluminosilicate
1 (bulk)	#3	0	0	4.23	9.17	0	8.51	0.04	0.83	ND	1.33	0.19	20.34	9.22	53.84	Quartz and aluminosilicate
3 (bulk)	#3	0.03	0.32	21.92	33.99	2.67	1.11	1.08	4.22		7.85	0.67	0.27	8.21	82.34	
2 (baryte)	#3						17.81		0		0	0	52.60	0	70.40	
4 (baryte)	#3	0	0	0			27.83	0	0		0	0	56.26	0	84.09	

**Table 2 molecules-27-05205-t002:** ICP-MS results in mg·g^−1^.

Sample	P	S	Ca	Na	Mg	Al	Si	K	Ti	V	Mn	Fe	Cu	Zn	Ga	Sr	Sn	Sb	Ba	Hg	Pb
1	0.3	0.6	<LOD	34.6	0.8	32	317	5.2	0.86	0.02	0.02	2.0	0.08	0.01	0.02	0.02	0.44	0.01	0.3	<LOD	6
2	3.2	3.3	4.4	1.1	1.3	51	223	4.0	0.07	0.03	0.01	0.5	0.26	0.02	0.22	0.15	0.02	0.05	4.3	0.01	35
3	0.6	14.0	14.0	1.2	0.6	11	101	0.9	0.04	0.03	0.03	4.0	0.13	0.10	4.84	3.52	0.17	0.10	86.0	0.43	232

## Data Availability

The data presented in this study are available on request from the corresponding author. The data are not publicly available due to the privacy policy of the author’s institution.

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
