# Peer review of "Secret Recipe Revealed: Chemical Evaluation of Raw Colouring Mixtures from Early 19th Century Moravia"

_molecules, 2022, doi:10.3390/molecules27165205_

Round 1

Reviewer 1 Report

This study aims to identify both organic and inorganic constituents in raw faience materials. The authors have conducted a very thorough and extensive study using several highly complicated characterization techniques. The results have demonstrated for the first time the possible type of organics in unfired samples. This would proof to be extremely useful in discovering such secret recipe. There are a few suggestions for this manuscript.

1. The title should include "pink powder" as the results contain this part too. 

2. The EDS (Table 1) does not show the element Sn (tin) compared to the ICP-MS in Table 2. Why?

3. For red glazes, Cu2O must be fired in a reduction atmosphere. Are there any evidences about the kiln's type commonly used in the area? Wood firing would be a good example.

4. If possible but not required, X-ray Diffraction might be good supplementary to further identify specific inorganic phases.

Author Response

Dear Editor,

I would like to thank reviewer for his excellent work that help to improve submitted article.

Reviewers comments:

This study aims to identify both organic and inorganic constituents in raw faience materials. The authors have conducted a very thorough and extensive study using several highly complicated characterization techniques. The results have demonstrated for the first time the possible type of organics in unfired samples. This would proof to be extremely useful in discovering such secret recipe. There are a few suggestions for this manuscript.

Comment 1:

The title should include "pink powder" as the results contain this part too. 

Answer:

Thank you for this comment. The title was changed to “Secret recipe revealed: Chemical evaluation of raw colouring mixtures from early 19th century Moravia”.

Comment 2:

The EDS (Table 1) does not show the element Sn (tin) compared to the ICP-MS in Table 2. Why?

Answer:

The tin was not detected by EDS technique. Based on this fact, the tin is not shown in Table 2.

Comment 3:

For red glazes, Cu2O must be fired in a reduction atmosphere. Are there any evidences about the kiln's type commonly used in the area? Wood firing would be a good example.

Answer:

There are several original technical sheets of historical kiln’s. There is also information about wood firing and heating of ceramic materials in (probably) closed chambers. Those chambers could be found in different high. However, there is no information about construction of ventilation systems. At this point, we are not able to say what kind of glaze was sintering in reduction and what in oxidation atmosphere.

Comment 4:

If possible but not required, X-ray Diffraction might be good supplementary to further identify specific inorganic phases.

Answer:

Authors agree that XRD is good technique to identify specific inorganic phases. Unfortunately this instrumentation is not available in authors laboratories and we do not plan to add this technique in submitted article.

Reviewer 2 Report

I have a couple somewhat substantial arguments against this paper as it is currently written; however, that could be completely changed just by altering the title and a few key words within the manuscript.

Finding a vial of powder in a ceramicist's workshop does not automatically mean that that material was used in glaze -- nor is it responsible for the same color in a ceramic glaze. Are shards of blue pottery present that show identical elemental compositions to the powder? Is that argument made? If not, you cannot argue that it was used in pottery glazes in central europe.

Additionally, prussian blue EXPLICITLY does not survive the sintering required for glaze melt to occur. The ferrocyanide bonds break and it disassembles (and likely oxidizes) into Fe3+, upon which it is surrounded by silicates/lead in a crystal field. If reduction-fired, with limited access to oxygen, it MAY form Fe2+, in crystal field, which does result in a sky blue color -- which is not prussian blue.

If anything, I would guess that it is a pigment used to paint/decorate the surface of ceramics after they were fired -- or the potter desperately tried to make it capture the blue color, could not do so, and was left with a vial of it in his workshop as the ferroferricyanide kept decomposing upon sintering.

There is a reason that this wasn't used and has never been used in ceramics.

Finding a single vial in a workshop of someone who tried to use it 300 years ago doesn't mean that it was successfully used in pottery in any capacity.

HOWEVER, all of these aspects could be addressed AND the title could state that it was found among a potter's pigments -- likely to decorate the surface of ceramics after sintering, and that would be fine.

Also, the paragraph: "One of the most common red pigments used in glazes/faience was 119
hematite α-Fe2O3 (or maghemite γ-Fe2O3) which causes red to brown shades depending on the conditions during firing.
120
The red hue was achieved under oxidizing conditions at a temperature higher than 700 °C [29]. Red glazes were also
121
achieved by use of other minerals, for example, cuprite Cu2O, crocoite (PbCrO4), or red lead (Pb3O4) [2931]. Based on
122
the results of the ICP-MS analysis, we hypothesize that the colouring agent in the pink powder was Fe or Cu found in
123
concentrations of 0.50 and 0.26 mg·g-1, respectively (probably in the form of hematite and oxide of copper Cu2O). When
124
copper oxides were used to obtain red colour, reduction conditions in the kiln were crucial because the final colour is
125
dependent on the presence of other oxidation states of copper oxides (Cu0, CuI, CuII). The presence of CuII is undesirable
126
in red glazes and thus the reduction conditions had to be optimised to obtain combination of precipitated Cu0 and Cu2O
127
only in the final glaze, which causes a distinctive red hue."

Is overwhelmingly false. Most copper reds are from nanoparticle plasmon resonance of copper-0 atoms between about 5-20 nm in diameter. This paragraph is misleading about that relatively well studied phenomena. Cuprite is NOT a common cause of copper reds in ceramic glazes/faience. There are multiple papers about particle size, size distribution, and color clarity as a function of these parameters.

Please rewrite/reword/readdress all of these points, and it could be a paper about prussian blue found among ceramic supplies -- for post-firing decoration, or possibly as a failed ingredient during pigment testing.

Author Response

Dear Editor,

I would like to thank reviewer for his excellent work that help to improve submitted article.

Rewievers comments:

I have a couple somewhat substantial arguments against this paper as it is currently written; however, that could be completely changed just by altering the title and a few key words within the manuscript.

Comment 1:

Finding a vial of powder in a ceramicist's workshop does not automatically mean that that material was used in glaze -- nor is it responsible for the same color in a ceramic glaze. Are shards of blue pottery present that show identical elemental compositions to the powder? Is that argument made? If not, you cannot argue that it was used in pottery glazes in central europe.

Answer:

Authors agree. The title was changed to “Secret recipe revealed: Chemical evaluation of raw colouring mixtures from early 19th century Moravia”.

Comment 2

Additionally, prussian blue EXPLICITLY does not survive the sintering required for glaze melt to occur. The ferrocyanide bonds break and it disassembles (and likely oxidizes) into Fe3+, upon which it is surrounded by silicates/lead in a crystal field. If reduction-fired, with limited access to oxygen, it MAY form Fe2+, in crystal field, which does result in a sky blue color -- which is not prussian blue.

If anything, I would guess that it is a pigment used to paint/decorate the surface of ceramics after they were fired -- or the potter desperately tried to make it capture the blue color, could not do so, and was left with a vial of it in his workshop as the ferroferricyanide kept decomposing upon sintering.  There is a reason that this wasn't used and has never been used in ceramics. Finding a single vial in a workshop of someone who tried to use it 300 years ago doesn't mean that it was successfully used in pottery in any capacity.

HOWEVER, all of these aspects could be addressed AND the title could state that it was found among a potter's pigments -- likely to decorate the surface of ceramics after sintering, and that would be fine.

Answer:

It is really good point. The Prussian blue should be totally damaged after sintering, so it is not PB anymore. We mentioned this point in text (see line 181-189) and we also discussed the utilization of blue powder for decorative purposes after the second firing.

Comment 4:

Also, the paragraph: "One of the most common red pigments used in glazes/faience was hematite α-Fe2O3 (or maghemite γ-Fe2O3) which causes red to brown shades depending on the conditions during firing.  The red hue was achieved under oxidizing conditions at a temperature higher than 700 °C [29]. Red glazes were also  achieved by use of other minerals, for example, cuprite Cu2O, crocoite (PbCrO4), or red lead (Pb3O4) [29–31]. Based on the results of the ICP-MS analysis, we hypothesize that the colouring agent in the pink powder was Fe or Cu found in concentrations of 0.50 and 0.26 mg·g-1, respectively (probably in the form of hematite and oxide of copper Cu2O). When copper oxides were used to obtain red colour, reduction conditions in the kiln were crucial because the final colour is  dependent on the presence of other oxidation states of copper oxides (Cu0, CuI, CuII). The presence of CuII is undesirable in red glazes and thus the reduction conditions had to be optimised to obtain combination of precipitated Cu0 and Cu2O only in the final glaze, which causes a distinctive red hue." Is overwhelmingly false. Most copper reds are from nanoparticle plasmon resonance of copper-0 atoms between about 5-20 nm in diameter. This paragraph is misleading about that relatively well studied phenomena. Cuprite is NOT a common cause of copper reds in ceramic glazes/faience. There are multiple papers about particle size, size distribution, and color clarity as a function of these parameters.

Please rewrite/reword/readdress all of these points, and it could be a paper about prussian blue found among ceramic supplies -- for post-firing decoration, or possibly as a failed ingredient during pigment testing.

Answer:

Thank you for this comment. This part was totally changed according your comment (see line 123-126).  

Round 2

Reviewer 2 Report

I am on board with the current version, less the "secret recipe revealed" part of the title.

I do not think that this is a secret recipe of any kind, but has valid findings with respect to xrf analysis of prussian blue alongside a potters other tools.

A more appropriate title may be, "Chemical evaluation of raw colouring mixtures from early 19th century Moravia, including Prussian Blue"